# Quantum Biochemistry Screening and In Vitro Evaluation of *Leishmania* Metalloproteinase Inhibitors

**DOI:** 10.3390/ijms23158553

**Published:** 2022-08-02

**Authors:** Cláudia Jassica Gonçalves Moreno, Henriqueta Monalisa Farias, Rafael de Lima Medeiros, Talita Katiane de Brito Pinto, Johny Wysllas de Freitas Oliveira, Francimar Lopes de Sousa, Mayara Jane Campos de Medeiros, Bruno Amorim-Carmo, Gabriela Santos-Gomes, Daniel de Lima Pontes, Hugo Alexandre Oliveira Rocha, Nilton Fereira Frazão, Marcelo Sousa Silva

**Affiliations:** 1Laboratory of Immunoparasitology, Department of Clinical and Toxicological Analysis, Health Sciences Center, Federal University of Rio Grande do Norte, Natal 59012-570, Brazil; claudia.mrn1@gmail.com (C.J.G.M.); johny3355@hotmail.com (J.W.d.F.O.); 2Postgraduate Program in Pharmaceutical Sciences, Health Sciences Center, Federal University of Rio Grande do Norte, Natal 59012-570, Brazil; bruno_portilo@hotmail.com; 3Postgraduate Program in Biochemistry, Biosciences Center, Federal University of Rio Grande do Norte, Natal 59012-570, Brazil; hugo-alexandre@uol.com.br; 4Academic Unit of Physics, Mathematics of the Education and Health Center, Federal University of Campina Grande, Campina Grande 58428-830, Brazil; monalisa_miller@hotmail.com (H.M.F.); rafaelfufcg@gmail.com (R.d.L.M.); niltonfrazao@gmail.com (N.F.F.); 5Postgraduate Program in Health Sciences, Health Sciences Center, Federal University of Rio Grande do Norte, Natal 59012-570, Brazil; talitakatiane@hotmail.com; 6Laboratory of Chemistry of Coordination and Polymers (LQCPol), Institute of Chemistry Chemistry Institute, Federal University of Rio Grande do Norte, Natal 59012-570, Brazil; francimar_junnior@hotmail.com (F.L.d.S.J.); mayarajane20049@hotmail.com (M.J.C.d.M.); pontesdl@yahoo.com (D.d.L.P.); 7Global Health and Tropical Medicine, GHTM, Institute of Hygiene and Tropical Medicine, IHMT, NOVA University of Lisbon—UNL, 1349-008 Lisbon, Portugal; santosgomes@ihmt.unl.pt

**Keywords:** *Leishmania* spp., leishmanolysin, metalloprotease inhibitors, MFCC (Molecular Fractionation with Conjugate Caps)

## Abstract

Leishmanolysin, also known as major promastigote protease (PSP) or gp63, is the most abundant surface glycoprotein of *Leishmania* spp., and has been extensively studied and recognized as the main parasite virulence factor. Characterized as a metalloprotease, gp63 can be powerfully inactivated in the presence of a metal chelator. In this study, we first used the structural parameters of a 7-hydroxycoumarin derivative, L1 compound, to evaluate the theoretical–computational experiments against gp63, comparing it with an available metal chelator already described. The methodology followed was (i) analysis of the three-dimensional structure of gp63 as well as its active site, and searching the literature and molecular databases for possible inhibitors; (ii) molecular docking simulations and investigation of the interactions in the generated protein–ligand complexes; and (iii) the individual energy of the gp63 amino acids that interacted most with the ligands of interest was quantified by ab initio calculations using Molecular Fraction with Conjugated Caps (MFCC). MFCC still allowed the final quantum balance calculations of the protein interaction to be obtained with each inhibitor candidate binder. L1 obtained the best energy quantum balance result with −2 eV, followed by DETC (−1.4 eV), doxycycline (−1.3 eV), and 4-terpineol (−0.6 eV), and showed evidence of covalent binding in the enzyme active site. In vitro experiments confirmed L1 as highly effective against *L. amazonensis* parasites. The compound also exhibited a low cytotoxicity profile against mammalian RAW and 3T3 cells lines, presenting a selective index of 149.19 and 380.64 µM, respectively. L1 induced promastigote forms’ death by necrosis and the ultrastructural analysis revealed disruption in membrane integrity. Furthermore, leakage of the contents and destruction of the parasite were confirmed by Spectroscopy Dispersion analysis. These results together suggested L1 has a potential effect against *L. amazonensis,* the etiologic agent of diffuse leishmaniasis, and the only one that currently does not have a satisfactory treatment.

## 1. Introduction

Leishmaniasis is a vector-borne disease caused by infection with several species of the *Leishmania* protozoa parasite. The disease affects people in tropical and sub-tropical areas worldwide, resulting in disability and morbidity [1]. Clinical manifestation of leishmaniasis can be subdivided into cutaneous (CL), mucocutaneous (MCL), and visceral (VL) forms. CL, the most frequent clinical form, can progress to be cured in most cases, whereas other patients can present debilitating lesions [2]. Some species of *Leishmania*, as is the case of *L. amazonensis*, cause non-healing cutaneous lesions and are responsible for the incurable form of human leishmaniasis known as diffuse cutaneous leishmaniasis [3].

It is a complex disease, where a diversity of parasites (at least twenty species of *Leishmania* are involved in the transmission), rapid environmental changes, and human activities represent the main challenges to controlling leishmaniasis [1,4,5]. These aspects are difficult for control strategies, and currently the vector and animal reservoir programs have not been reliable or effective [6,7,8]. Moreover, there is no vaccine against human leishmaniasis and chemotherapy regimens are highly toxic and expensive [1].

*Leishmania* species have a remarkable ability to subvert or modulate innate and adaptive immune responses of the vertebrate host [9,10]. This protozoan adopted a sophisticated mechanism that enables it to avoid the hostile defense system and survive as an obligate intracellular parasite [11]. In that order, *Leishmania* parasites encode a major surface protease, a 63-kDa glycoprotein (gp63), also named leishmanolysin (EC 3.4.24.36). The glycoprotein is a metalloproteinase, Zn^2+^ dependent, with the identical characteristics of mammalian metalloproteinases (MMPs) [12,13], and represents the main express membrane protein [14]. 

Studies have recently revised the gp63 function as mainly a *Leishmania* virulence factor [9,10,14,15,16,17]. Therefore, gp63 can play a role, such as the degradation of the extracellular matrix and proteolytic activity [15]. Regarding this last one, the ability to degrade different peptides and protein substrates, including, collagen, fibrinogen, and hemoglobin, makes the macrophage extracellular matrix susceptible to gp63 hydrolysis, which favors parasites migrating inside and dissemination [16,17]. Additionally, gp63’s effect on antimicrobial and inflammatory functions of macrophages has been extensively documented [16]. The protein can be released as an intracellular parasite survival factor via exosomes, and negative regulate host cell signaling. Subsequently, the important suppression of immune cells reinforces autophagy [9,10]. As reviewed, an in vivo study demonstrated that gp63 delivered by *Leishmania* spp. exosomes resulted in lowering of serum cholesterol and enhancement of murine liver infection. On the other hand, *L. amazonensis* promastigotes deficient in gp63 were more susceptible to human histones (H2B), with an anti-leishmania property [15,16].

In the last decades, proteases have been considered promising targets in protozoa infection treatment. Due to their remarkable key roles during infection, studies indicate protease inhibitors as an alternative to therapy against drug-resistant parasites [18]. Thus, a few compounds have been used as metalloproteinases inhibitors in *Leishmania major* promastigote cultures, but such compounds have not demonstrated the expected cytotoxic effect [19,20]. Furthermore, the development and molecular characterization of higher-affinity metalloprotease inhibitors may provide a novel alternative for leishmaniasis treatment.

Consequently, this study aimed to use quantum biochemistry characterization to evaluate a specific metalloprotease inhibitor molecule designed and synthesized in the Chemistry Institute of UFRN, Brazil, followed by in vitro experiments for an evaluation of the potential inhibitors selected.

## 2. Results

### 2.1. Identification of Potential Metalloprotease Inhibitors

In silico prediction performance was evaluated through calculations of the binding free energy, for the binding of compounds to gp63 using a non-physical thermodynamic cycle. Furthermore, for the level of accuracy, molecule size was taken in consideration during the prediction calculation.

For all ligands, a binding pose that captures the main features of their interactions with gp63 was among the negative results (see Appendix A for the total number of docked complexes obtained for each compound by the scoring function). As related to other ligands, L1 showed the lowest EFE (−8.07 kcal/mol) and FIE (−9.86 kcal/mol) values. These interaction energies displayed by the ligands anchored to the glycoprotein seem to reflect a strong and thermodynamically favorable interaction. Thus, docking identified the pose closest to the crystallographic structure of gp63, noticing one ligand that has less energy as being the most favorable of all ligands.

The top-ranked pose with the lowest docked binding affinities and high docking scores are generally used as a standard selection in most docking programs (Table 1). Potential calculations in the binding poses for each ligand are presented in Figure 1. DETC has the most favorable pose (−4.45 kcal/mol) with established form linkages between the amino acids of greater proximity. Different sets of hydrogen-bonding interactions with the chain residues of LEU357 and LYS307 were observed. In addition to the hydrogen–carbon with amino acids GLU376 and GLN311, finally a π-alkyl bond (covalent bond) with VAL377, ALA375m and ALA356 was seen in the complexes selected by molecular docking (Appendix A).

For DOXY, the binding affinity was found to be −5.46 kcal/mol. The orientation and hydrogen bonding of DOXY can be seen within the gp63 active site (Figure 1). It was possible to identify the hydrogen bonding with the amino acids ALA356, GLU376, ALA375, and PHE372; also, hydrogen–carbon bonds with the amino acids THR355 and VAL377 and a π-alkyl bond with LEU537 (Appendix A). The smallest ligands, terpinen-4-ol (TERP-4), presented −5.04 kcal/mol and mediated the hydrogen bond interactions between the amino acids of greatest proximity, as is the case of Asp288, Gly291, Val289, and Arg290, and can establish a covalent bond (π bonds type) with Glu321, Leu389, Val286, and Leu389 (Appendix A).

The ligand L1 was mediating the top-ranked pose in this study. π-Alkyl bonds were identified with the amino acids ALA346, VAL 261, and LEU224. π-Lone Pair bonding, with an amine side chain of PRO 347, also had a side chain of LEU344. Furthermore, non-covalent interaction π-π stacked type was established with the aromatic rings of HIS264 from the catalytic center of gp63. A covalent interaction, type π-sigma bond, had the amino acids ALA349 and carbon-hydrogen bond with side chain residues of GLU265 (Appendix A). The neutral behavior represents negative values of the interpolated charges of the direct protein–ligand interaction. DETC, DOXY, and L1 ligands results shown the electrostatic cloud of the receptor involving the binding and the anchorage site is located in a region of neutral interaction, varying to negative in regions of strong interaction (Figure 1). TERP-4 showed that, for the electrostatic cloud, it is possible to notice it in a neutral region of interaction.

Still using the Discovery Studio software, it was possible to analyze the electrostatic cloud of the gp63 and the docking point seems to be located in a region of neutral interaction (Figure 1). It was also possible to observe a very electronegative region close to the interaction site with side amine residues of Glu376. In addition to being thermodynamically favorable for interaction, the connection is located inside the active site of gp63.

### 2.2. Quantum Energy Calculation between the Ligands and gp63 Protein

A three-dimensional model was used to carry out the quantum energy calculations. The energy calculations between the ligands and the amino acids of gp63 were performed using the MFCC (Molecular Fractionation with Conjugate Caps) technique, which enables calculations involving proteins, biological macromolecules, and DNA, focusing on providing accurate information on the interaction energy between the complexes [21].

It was possible to establish the specific amount of binding energy between each glycoprotein amino acid residue within a 12 Å selected radius and ranked from the most relevant interactions pointed out previously by the Discovery Studio (DS) software. In the present study, the binding energy values between gp63 and the ligands were measured by quantum mechanical calculations in the scope of the Density Functional Theory (DFT), using the MFCC scheme as the mechanism of attraction and repulsion. Thus, the Local Density Approach (LDA) was used to determine the exchange-correlation potential for the protease, due to the active site cavity in which a structural water molecule plays an important role. This water molecule is explicitly included in the calculations because it participates directly in the binding interactions [22,23].

A BIRD graphic panel shows the result obtained by the MFCC calculation of the relevant gp63 amino acid residues at the binding region with each ligand (Figure 2). DETC indicated positive and strong negative energies among the eight amino acid residues (Val377, Met359, Lys307, Leu357, Gln311. Ala375, Glu376, and Ala356) evaluated (Figure 2A). A comparative analysis between the energy values with distances calculated from the geometric center of both of the amino acid residues demonstrated a stronger interaction of DETC with the Lys307 residue (−13.8 kcal/mol and 8.02 Å distance).

The Gln311, Glu376, Ala356, and Leu357 residues exhibited interaction energy inflows of −8.2, −6.2, −4.1, and −3.5 kcal/mol and 5.00, 5.02, 5.04, and 6.84 Å, respectively. However, the amino acid residues Val377 (0.0 kcal/mol and 4.84 Å), Ala375 (1.2 kcal /mol and 6.81 Å), and Met359 (2.6 kcal/mol and 9.09 Å) contributed with null or a positive interaction energy.

The following amino acid residues, namely, Ala356, Ala375, Glu310, Leu357, Lys307, Pro379, and Thr355, established strong attractive bonds with the DOXY ligand and also repulsive bonds, as is the case of Phe372, Glu376, and Ala306 (Figure 2B). The residue that showed the highest negative interaction energy was Glu310 (−18.6 kcal/ mol and 8.42 Å), and Ala356 (6.0 kcal/mol and 8.13 Å) presented a higher repulsive interaction energy (Figure 2B).

The smallest ligand, terpinem-4-ol, showed predominantly attractive interactions with the amino acid residues Val289, Val286, Tyr319, Thr390, Pro287, Gly291, Asn288, and Arg290 (Figure 2C). The residue Arg290 showed a stronger binding energy (−11.83 kcal/mol and 6.39 Å). In turn, Val289 (6.15 kcal/mol and 4.55 Å) exhibited a repulsive energy.

The L1 ligand was the only one that was able to interact strongly with the gp63. Furthermore, all the amino acid residues Ala346, Ala349, Glu265, His264 Leu224, Leu344, Met345, Pro347, and Val261 showed a predominant attractive interaction with the compound (Figure 2D). An important observation is that the amino acid residues are subject to the negative interaction energy with the main contribution to the total L1-gp63 binding energy in the decreasing sequence Glu265 > His264 > Ala349 > Leu344 > Pro347 > Leu224 > Val289 > Ala346 > Met345 (Figure 2D). The total contribution of binding energy to each ligand follows an increasing order TERP-4 < DOXY < DETC < L1 (Figure 3). A binding energy value of less than one indicates a favorable covalent bond type.

The present quantum study provides important information about the specific protein–ligand interaction, in addition to being useful in planning future protein inhibitors. It is important to note that the present study emphasizes the analysis of the specific binding interaction between the ligament filaments and the protein component, in which the calculation is based on experimental crystalline structures.

### 2.3. L1 Molecule as a Potential Inhibitor against Leishmania amazonensis

In vitro antiparasitic activity of in the silico ranked compounds DETC, DOXY, and L1 was evaluated against promastigote forms of *L. amazonensis*. Results showed IC_50_ values of 1.24 ± 0.08, 4.97 ± 0.011, and 26.82 ± 0.1 µM, respectively (Table 2). AmpB was used as positive control and showed an IC_50_ value of 0.906 ± 0.06 µM.

In this study, we investigated whether the compounds tested in vitro promoted changes in the parasite membrane or induced a necrosis mechanism against *L. amazonensis* promastigotes. Subsequently, the flow cytometry analysis of untreated promastigote forms revealed low staining (negative control—Ctrl). About 0.31% and 5.30% of the parasites were stained with PI and Annexin V, respectively (Figure 4).

Annexin-positive cells were observed in Amphotericin B (99.3%) and DETC (97.1%) after 24 h of treatment with 100× the IC_50_ value (124 µM). The results are characteristic of late apoptosis or cell death (Appendix A). However, there was a considerable increase in PI-positive cells (57%) in parasites treated with L1. In the cells treated with DOXY, 45.6% were also in the process of necrosis by PI staining (Appendix A).

### 2.4. Morphological and Structural Changes of Leishmania amazonensis Promastigotes in the Presence of Potential Inhibitors

The effect of the tested compounds in *L. amazonensis* promastigote morphology was elucidated 24 h after incubation through electron microscopy techniques [24,25]. *L. amazonensis* promastigotes were treated with the compounds at IC_50_ concentrations (1.24 µM). Untreated parasites were used as control. *L. amazonensis* promastigotes treated with the compound L1 and DOXY exhibited cell surface projections, pores, and flagella alteration (Figure 5 and Appendix A). Cell surface discontinuities, shrinkage of the parasite surface caused by DETC, and cytoplasmic loss, as well as possible disorganization of the cytoskeleton, can be observed (Appendix A). The extensive morphological changes observed can lead to parasite destruction. The chemical elements present in the cellular content, observed on the SEM images, were characterized by Energy Dispersive X-ray Spectroscopy (EDS). Thus, higher organic contents were observed in *L. amazonensis* promastigotes treated with L1 by marking the carbon (C, red) and silicon (Si, green). The EDS curve shows a peak close to 2.4 keV corresponding to carbon (C) and a peak displacement of the silicon standard (1.8 keV) for both compounds (Figure 6).

### 2.5. L1 Molecule Demonstrated a Well-Tolerated Profile against the Cells Lines 

The cytotoxic effect of the L1 molecule was assessed by the MTT method on RAW and 3T3 cells (Figure 7). In both cell lines treated with L1 for 24 h, a concentration-dependent effect was observed. The CC_50_ values of 472 µM were found for RAW cells and 185 µM for the 3T3 culture. According to the results obtained, compound L1 has a low toxicity at concentrations below 478 µM and 185 μM for the 3T3 and RAW cell lines, respectively. The selectivity index (SI) of L1 was calculated, taking into consideration the CC_50_ values obtained in mammalian cell lines and the parasite IC_50_ (Table 3). The L1 compound had a 149- and 380-times greater toxicity for the parasite than for the mammalian cell, pointing towards a high selectivity for parasite cells.

## 3. Discussion

The main control strategy against human leishmaniasis consists of treatment with drugs, despite their toxicity and cost problems, and besides an increased number of resistance notifications [1,4]. Consequently, it is necessary to search for compounds that are more effective and less toxic [26]. As an alternative, treatment improvement through the identification of new targets, both in parasites and in host cells, is required [27,28]. In this sense, we firstly performed in silico screening of an attractive target for an important glycoprotein (gp63) present in trypanosomatids.

Molecular modeling of biomolecules allows not only the prediction and refining of three-dimensional structures but also the correlation of structures with their properties and functions [29]. Protein–ligand docking is performed by modeling the binding affinity between the ligand and protein and the structure of the protein–ligand complex using various criteria, such as if the geometry of the pair is complementary and integrating the favorable biochemical interactions of the MFCC [22,30].

Proteins are structurally differentiated by their side chain groups with different chemical reactivities and physicochemical properties [31]. The side-chain groups include (i) hydrogen groups, (glycine); (ii) alkyl groups—for example, alanine (ALA), valine (VAL), and leucine (L) form hydrophobic interactions; and (iii) phenylalanine (PHE) and tyrosine (THR), which form bonds mainly to planar molecules [29,31]. DETC, DOXY, TERP-4, and L1 ligands presented bonds of the π -π, π-alkyl type, hydrogen bonds, and final molecular binding energy, negative with gp63.

The compound 4-({bis [(pyrydin-2-yl) methyl] amino} methyl) -7-hydroxy-2H-1-benzopyran-2-one, named L1, was synthesized at the UFRN Chemistry Institute and its structure was provided to carry out the in silico analyses. Our data showed the interaction between L1 and gp63 was due to their structure and size. Covalent bonds with the amino acids—alkyl group and hydrogen bonds were observed. A non-covalent bond (π-π stacking) type also was observed between the amino acid residues and the aromatic rings present on L1. The compound showed the lowest final intermolecular energy, compared to the others studied compounds.

Structurally, the compounds L1 > DOXY > DETC > TRP-4 can act as Lewis bases by coordinating with the Zn^2+^ ion of the catalytic center of gp63. Additionally, molecules such as DETC, L1, and DOXY (Table 2) are chelating agents, since they have more than one donor atom capable of coordinating with the metallic center at the same time, such as sulfur (S), nitrogen (N), or oxygen (O) atoms forming particularly stable complexes.

DETC was selected for this study due to the potential *Leishmania* spp. inhibitor of superoxide dismutase (EC 1.15.11) [32]. These metalloenzymes participate in the elimination of superoxide radicals (O^2−^) from both mammalian cells and pathogens, including trypanosomatids, preventing them from reactive oxygen species [33]. Here we simulated the overall DETC binding contributions and intermolecular interactions with a metalloprotease. Doxycycline has a great identified property [34]; however, due to its relatively large size in comparison with the other compounds, it can cause a steric impediment, inducing direct interaction with the metallic center of Zn (II) [30].

Using the three-dimensional model to perform quantum energy determination calculations, it was possible to establish the specific amount of binding energy between each glycoprotein amino acid (gp63) within an adopted 12 Å cutting radius and selected the amino acids with greater interaction as pointed out by the Discovery Studio program (DS). The present values were measured by calculations of quantum mechanics between gp63 and the ligands in the scope of the Density Functional Theory (DFT) using the technique of Molecular Fractionation of Conjugated Caps (MFCC). For this work, the Local Density Approach (LDA) was used to determine the exchange-correlation potential for the protease, as it is known that the active site is a closed cavity in which a structural water molecule plays an important role. This water molecule is explicitly included in the calculations because it participates directly in the binding interactions [35].

Although, in the evaluation of the interactions using quantum methods, each significant amino acid can influence the mechanism of attraction and repulsion. Thus, the present quantum study provides important information about the specific protein–ligand interaction, in addition to being useful in planning future protein inhibitors. It is important to note that the present study emphasizes the analysis of the specific binding interaction between the ligament filaments and the protein component, in which the calculation is based on experimental crystalline structures [31]. Complete calculations of the quantum mechanics of the MFCC type were performed to investigate the binding mechanism of four possible *Leishmania* glycoprotein inhibitors. Thus, the results of this work show promise, indicating that the four compounds (DETC, DOXY, TERP-4, and L1) point to the potential for inhibition of the gp63 protein due to the high negativity, which may indicate less energy loss during the union of these molecules.

Subsequently, the in vitro study evaluated the effects of the three compounds that demonstrate, in silico, covalent bonds with the amino acids—alkyl group and hydrogen bonding between DETC, DOXY, and L1 with *Leishmania* virulent factor gp63. Thus, promastigote forms of *Leishmania amazonensis* were used and all compounds showed anti-*Leishmania* activity. L1 induced changes in the membrane and morphology of the protozoan, and consequently the death of the parasite. L1 triggered cell disorganization and inhibition of the parasite by the mechanism of cell necrosis, a mechanism similar to that observed in cells treated with the DOXY compound.

DOXY is an antibiotic with a wide clinical spectrum, including being antiparasitic. Clinical studies have shown complete clinical regression of cutaneous leishmaniasis lesions after oral administration of doxycycline [36]. The DETC compound was able to increase the inhibition of the parasite in vitro and decrease the size of the lesion and the parasitic load in the in vivo model [32,33]. Thus, its probable inhibitory capacity is due to its performance as a metal chelator. The data obtained showed DETC as a promising compound, as it significantly inhibited the growth of *L. amazonensis*, corroborating in silico data.

The shrinkage and morphology change in promastigote forms of *L. donovani* has been observed in previous studies using Miltefosine [37,38]. Together, the damage leads to DNA fragmentation, loss of mitochondrial membrane potential, which suggestive of an apoptosis process, and necrosis, as observed by flow cytometry. These results detected by scanning microscopy may be indicative of membrane damage by factors such as drug-induced oxidative stress [24,25]. Microscopy clarifies both the cellular biology of the protozoan as well as the result of chemotherapy since the parasitic plasma membrane frequently exhibits morphological changes in response to microbicide drugs [24].

However, microscopy associated with the spectroscopic approach allows specific elementary mapping to be performed, as well as providing visible results that allow the determination of the mechanisms of action of such compounds [25]. Therefore, the validation of the externalized cellular content was done through the analysis of electron dispersion using the analytical technique of Energy Dispersive X-ray Spectroscopy (EDS); that is, the elementary analysis after SEM visualization of *L. amazonensis* promastigotes treated with the L1 molecule.

The L1 molecule showed characteristics of a promising compound and could be a candidate compound for a protease inhibitor, showing low cytotoxicity to mammalian cells and having a high selectivity index. Currently, the drugs used to control leishmaniasis have significant cytotoxicity, in addition to cases of drug resistance [1,7]. Thus, by using amastigote forms, although they have a lower number of metalloproteases, the protein is exposed to interaction with the host’s immune system [11,39]. In this sense, a future in vivo study is likely to verify this interaction and the coordination of L1. We intended to verify L1’s mechanism and pathway interference, as well as show its ability to treat *Leishmania-*infected animals.

## 4. Materials and Methods

### 4.1. Leishmania 63-kDa Glycoprotein Structures

The only crystalline structure available of glycoprotein 63 kDa (gp63) from *L. major* described the protein containing 478 amino acid residues, with predominantly β–sheet secondary structure and an N terminal domain [40]. The central and C-terminal domains were obtained from the PDB (Protein Data Bank) database using the 1LML access code. The secondary structure (Appendix A) was analyzed visually by Discovery Studio Software (BIOVIA discovery studio visualizer, 2014), as showed in Appendix A.

### 4.2. Ligand Structures

A literature review was carried out in order to identify the possible ligands that may show good computational results as a proteolytic inhibitor. The research was performed using the SciELO, PubMed, Lilacs, and Medline databases, between 2018 and 2021. Ligand structures were searched in small-molecule databases, such as the ZINC database (http://zinc.docking.org, accessed on 26 March 2019), PubChem (https://pubchem.ncbi.nlm.nih.gov, accessed on 8 November 2018), and Chemspider bank (http://www.chemspider.com, accessed on 25 January 2019). Files were generated in the .pdb extension and the structures of the compounds were visually analyzed by Visual Molecule Dynamics and Discovery Studio—VMD.

Four ligands were selected to conduct the in silico analysis. Among the selected compound, three of them were included in the current study: doxycycline (ZINC21984014), diethyldithiocarbamate (ZINC03633221), and terpinen-4-ol (ZINC03861537). Finally, the fourth compound was synthesized at the Chemistry Institute of the Federal University of Rio Grande do Norte, namely, 4-({bis[(pyridine-2-il)metil]amino}metil)-7-hidroxy2H-1-benzopyran-2-ona, identified as the L1 molecule and drawn using VMD software. In Appendix A contains more detail of the compounds.

### 4.3. Docking

The docking of the *Leishmania* glycoprotein or gp63 with the ligand was performed using the free software Autodock1.5.6 software (The Scripps Research Institute and Olson Laboratory [41]. The docking protocol employed the Lamarckian Genetic Algorithm (LGA) in a cubic grid, 126 × 126 × 126 Å, receiver-cantered. For the docking simulations, the protein was considered a rigid receptor while the ligands were considered flexible. Finally, ten complexes or conformations of each protein–ligand interaction were obtained. The best scoring one presented within each cluster was used for the free-energy calculations. Two parameters were considered: the final intermolecular energy and the estimated binding free energy (kcal/mol). Both were estimated by the Autodock 1.5.6 software.

### 4.4. Molecular Fractionation Conjugate Caps (MFCC)

Toward the quantum, mechanical-based simulations were used, namely, Density Functional Theory, Local Density Approximation for the exchange-correlation functional, and a double numerical plus polarization basis set, using the software DMOL3 [42]. Interaction energy calculations were performed by using the Materials Studio software, employing the Dmol3 code, as well as the exchange and correlation function: Generalized Gradient Approximation (GGA) was calculated using the Molecular Fractionation Method with Conjugated Caps (MFCC).

To verify the distance between the interacting residues of the receptor–ligand, the software Biovia Discovery Studio was used, through the centroid monitors that mark the geometric center of the selected atoms. The cut-off radius assigned to the software was R = 12 Å. The optimization of the structures was done through the same software with the addition of hydrogens employing the option chemistry—hydrogens—add to close all the valence, as previously described [43].

The linker molecule was determined as L and the amino acid residue that interacts with the linker as Ri. The Ci-1 (Ci+1) represents the cap formed from the neighboring residues of residue R1 along the protein chain. The study included two of the closest amino acid fragments on each side of the Ri residue, which was used to construct the Ci-1 and Ci+1 cap. For these fragmented structures, the interaction energy between the ligand and the individual fragments was calculated according to the following equation proposed by Gordon [23]:E_Interaction_ (L↔Ri) = E (L ↔ C1-i Ri Ci + 1) − E (L ↔ Ci-1 Ri Ci + 1) − E (L ↔ Ci-1Ci + 1) + E (L ↔ Ci-1Ci + 1)
where the term EI (L − Ri) represents the total quantum energy of the system, which is schematically represented in Appendix A. The term E (L + Ci − i Ri + Ci) indicates the formed energy of the ligand plus the residue Ri next to the conjugated caps (F1). E (L + C1-i Ci + 1) represents the total energy of residues and the conjugated caps (F2); E (L ↔ Ci-1Ci + 1) is the total energy of the system formed by caps and ligand (F3). Finally, E (C1-i Ci + 1) indicates only the conjugated amino acid caps (F4). The interaction energy calculation EI (L − Ri) is expressed in kcal/mol.

### 4.5. Antiparasitic Activity of In Silico Selected Compounds 

#### 4.5.1. Compounds

Selected compounds, such as diethyldithiocarbamate (DETC), Amphotericin B (AmpB), doxycycline hyclate (DOXY), RPMI 1640 medium, and Resazurin sodium salt, were purchased from the Sigma Company (San Luis, Missouri, USA). All compounds were dissolved using ultrapure water to final concentrations. The L1 compound was synthesized in accordance with the procedure described in the literature [44]. A solution containing water and 10% of dimethyl sulfoxide—DMSO (Neon Comercial, SP, Brazil)—was used to dissolve L1, and the final concentration was diluted in RPMI 1640 medium. Resazurin sodium salt (3 mM solutions) was prepared and stored at 4 °C protected from light.

#### 4.5.2. Parasites

The promastigote form of *Leishmania amazonensis* (MHOM/BR/1973/M2269 strain) was maintained in RPMI 1640 medium at 26 ± 2 °C, supplemented with 10% heat-inactivated Fetal Bovine Serum (Gibco, ThermoFisher, Waltham, Massachusetts, USA), penicillin (Sigma Aldrich, San Luis, MO, EUA), and streptomycin (ThermoFisher, Waltham, MA, USA). All assays were performed after the promastigote culture form reached the exponential growth phase.

#### 4.5.3. Anti-Parasitic Activity Determined by Microscope Counting and Resazurin Assay 

The antiparasitic activity was evaluated by microscopy counting according to the flagellum movement and the resazurin dye was employed for promastigote susceptibility testing following the methodology described by Rolón and colleagues [45].

*L. amazonensis* promastigotes were incubated with DOXY, DETC, L1, and AmpB as a positive control. Assays were performed using sterile 96-well plates using late log-phase promastigotes (1.0 × 10^7^/mL) per well and drug concentrations ranging from 0.1 μM to 500 μM. After 24 h of incubation at 26 °C, 20 µL resazurin (3 mM in phosphate-buffered saline, PBS) was added, and the plates were incubated for a further 24 h. The assays were carried out in triplicates and the absorbance was read at 570 excitations and emission at 600 nm using a 96-well plate reader (Epoch Biotek, Winooski, VT, USA). Results were expressed as resazurin reduction to resorufin percentage of viable parasites compared to the untreated samples. Half-maximal inhibitory concentration (IC_50_) was calculated by applying sigmoidal regression to the dose–response curves [46].

#### 4.5.4. Analysis of Cell Membrane Integrity

Propidium iodide (PI) and Annexin V-FITC (fluorescein isothiocyanate) staining were used to investigate changes in the membrane of *L. amazonensis* promastigotes treated with AmpB (0.108 and 108 µM); DOXI (25.5 and 225 µM), DETC (4.43 and 443 µM), and L1 (1.24 and 124 µM) by flow cytometry. The assay was performed according to the manufacturer’s instructions of the Annexin V FITC Apoptosis Detection Kit (Invitrogen, BD Biosciences Inc, San Diego, CA, USA). In parallel, untreated parasites were used as a negative control. After 24 h of incubation, parasites were washed and the obtained pellet was washed with PBS by centrifugation at 439× *g* for 5 min at 4 °C and the supernatant was discarded. Subsequently, treated parasites were resuspended in the binding buffer and labeled with Annexin V and PI for 10 min, at room temperature, and under light protection. Then the parasites were analyzed by a flow cytometer (FACSCanto II, BD Biosciences, OR, USA) using FACS DIVA software, version 6.1.2 (Becton Dickson). A total of 20,000 events were considered for each experimental condition and three independent experiments were performed. The identification of the parasite subpopulations and the determination of its frequency were made using FlowJo software Version vX.07 1997-2014 (Tree Star, Ashland, OR, USA).

#### 4.5.5. Morphological Analysis Using Simultaneous Scanning Electron Microscopy (SEM) and Energy Dispersive X-ray Spectroscopy (EDS)

Treated *L. amazonensis* promastigotes (as described in Section 4.5.3) were collected and centrifuged at 439× *g*, for 10 min, and washed twice with sterile PBS. After the last centrifugation, the supernatant was discarded and the parasites were fixed to coverslips with 2.5% glutaraldehyde in 0.1 M sodium cacodylate buffer for 1 h at room temperature. After fixation, the samples were washed with the same buffer twice and post-fixed in 1% osmium tetroxide solution, 0.8% potassium ferrocyanide, and 5 mM calcium chloride, protected from light. Then, the parasites were washed again, and dehydrated in increasing concentrations of acetone (30, 50, 70, 90, and 100%), for 10 min each. After the dehydration, the samples were dried by the critical point method. The coverslips were mounted on the plating supports and preceded by gold metallization. The samples were analyzed by scanning electron microscopy under an FEG microscope (Model augira, Brand Zeiss, Oberkochen, WB, GER). In addition, a chemical microanalysis technique in conjunction with SEM was performed using Energy Dispersive X-ray Spectroscopy (EDS).

### 4.6. Cell Lines and Cell Culture

RAW 264.7 macrophages (ATCC number TIB-71) and 3T3 fibroblast (ATCC CRL-1658) cell lines were used for the cytotoxicity assay. Cells were grown in DMEM medium (Cutilab, Campinas, SP, BR) supplemented with 10% fetal bovine serum, 2 mM glutamine, streptomycin/penicillin (100 U/mL), and incubated at 37 °C in a 5% CO_2_ atmosphere.

Cytotoxicity Assay against 3T3 and Raw Cells Lines

Only one compound, L1, was evaluated. To the best of our knowledge, its cytotoxicity activity has not been described in the literature. The remaining compounds are drugs currently used in clinical practice.

3T3 and RAW cell viability was estimated by the ability to live cells to reduce the yellow dye 3-(4,5-dimethyl-2-thiazolyl)-2,5-diphenyl-2H-tetrazolium bromide (MTT, Sigma Chemical Co. St Louis, USA) to purple formazan crystals. For all experiments, 100 μL of cells were seeded in 96-well plates (5.0 × 10^3^ cells/mL). After 80% cell confluence, L1 was added to each well at concentrations ranging between 0 and 2390 µM, and the cells were incubated for 24 h. As a control, untreated cells were incubated with 10% DMSO. At the end of incubation, plates were centrifuged and the medium was replaced by a fresh medium (150 μL) containing 0.5 mg/mL MTT dissolved in DMEM. Formazan product was dissolved with 10% DMSO and absorbance was measured using a 96-well plate reader (Epoch Biotek, Winooski, VT, USA), with absorbance at 570 nm [47]. The drug effect was expressed as the percentage and the 50% cytotoxic concentration (CC_50_). The selectivity index (SI) was calculated through the ratio between the CC_50_ and IC_50_ of promastigotes from *L. amazonensis*.

### 4.7. Statistical Analysis

The experiments were carried out in triplicate and independently. The results are presented as the arithmetic mean and standard deviation (±). Data were analyzed using ANOVA followed by Dunnett’s post-test multiple comparison test. To determine the statistical significance between groups, a one-way test was used. Statistical analysis was performed using Prism 6.0 (GraphPad Software, San Diego, CA, USA).

## 5. Conclusions

There is no effective vaccine against human leishmaniasis and currently treatment has more disadvantages, such as variable efficacy and drug resistance. Consequently, it is necessary develop new approaches and search for compounds that are more effective and less toxic. Trypanosomatid parasites increase the expression of proteases, such as gp63, which is associated with immune system evasion strategies. Thus, proteases are considered promising targets in the treatment of protozoan infection. In fact, they could provide an alternative therapy against parasites, due to this pivotal role during infection.

In this study, we demonstrated the strong gp63 inhibition observed by Molecular Docking and Molecular Fractionation with Conjugate Caps (MFCC). Both computational analyses allowed the conclusion that all four compounds tested show high constants of free energy of the bonds and the intermolecular energy, which is probably thermodynamically favorable. Glycoprotein amino acid residues Lys307, Glu 310, Glu265, and Arg290 can be considered as the anchors responsible for the covalent interactions observed by the MFCC. These results can be used in the future for the improvement of design and as a synthesis of new, even more specific peptides against metalloprotease from *Leishmania* spp., as well as other trypanosomatid parasites.

The anti-parasitic activity enabled the validation in vitro of the selected in silico compounds, with the ability to bind to the 63-kDa glycoprotein (gp63). Therefore, the L1 compound showed the characteristics of a promising compound and a solid candidate compound for protease inhibitor, with low cytotoxicity for mammalian cells and a high selectivity index. Future analyses of intracellular amastigote form cytotoxicity and in vivo study may allow complete therapeutic characterization.

In addition, L1 studies can also be directed to combination therapy, being used as a protease inhibitor, as it is already done in other infections in which protease has important roles, such as those caused by viruses.

## Figures and Tables

**Figure 1 ijms-23-08553-f001:**
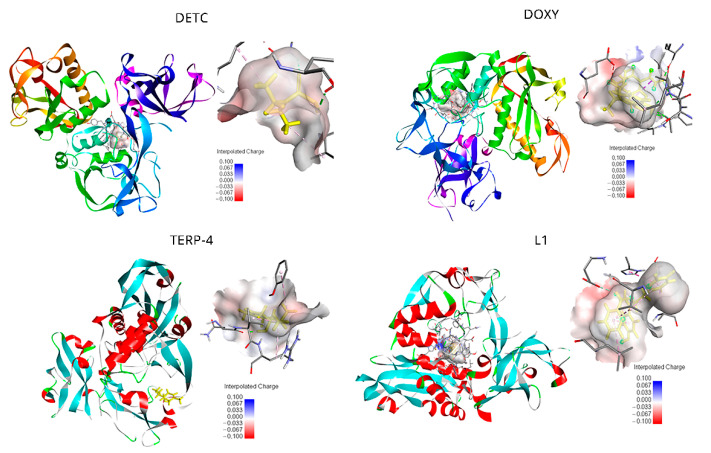
Active site of the *Leishmania major* glycoprotein of 63 kDa (gp63) in complex with diethyldithiocarbamate (DETC); doxycycline (DOXY); terpinen-4-ol (TERP-4); and 4-({bis[(pyridine-2-il)metil]amino}metil)-7-hidroxy2H-1-benzopyran-2 (L1). The binding interaction, forming an electrostatic cloud between the protein and ligand, is shown in grey and the docked ligands in yellow. Hydrogen bonds are depicted as green dashed lines, and the binding pocket surface is color-coded according to the interpolated charge (blue/red = positive/negative). The neutral behavior is notable and keeps diverging only by negative energy in sites of direct ligand interaction.

**Figure 2 ijms-23-08553-f002:**
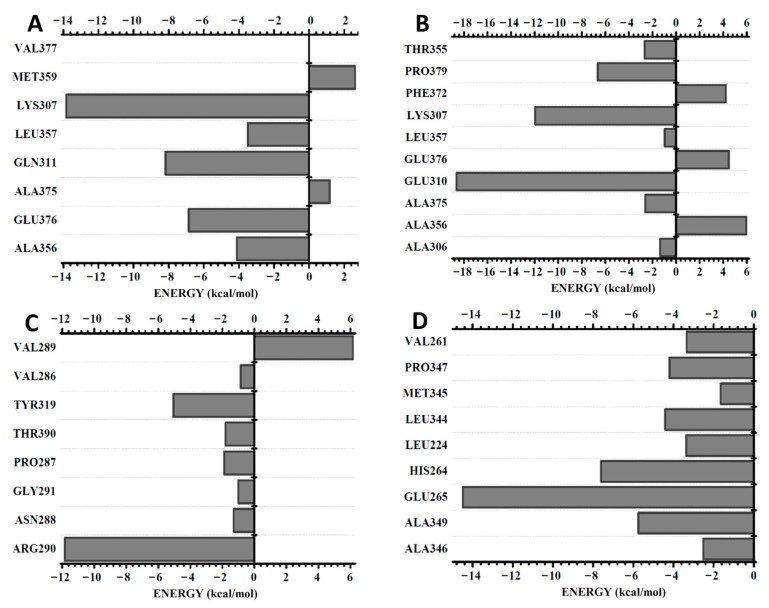
BIRD graphic panel showing the most relevant residues of the gp63 contributions that bind to the ligands: (**A**) DETC, (**B**) DOXY, (**C**) terpinen-4-ol, and (**D**) L1. Distances (Å) are indicated and binding energy is shown in kcal/mol.

**Figure 3 ijms-23-08553-f003:**
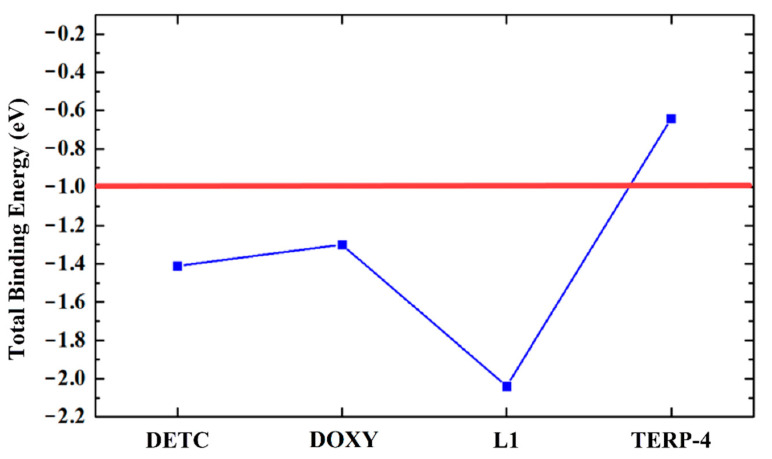
Graphical representation of the total binding energy (eV) of all ligands anchored with the gp63 protein.

**Figure 4 ijms-23-08553-f004:**
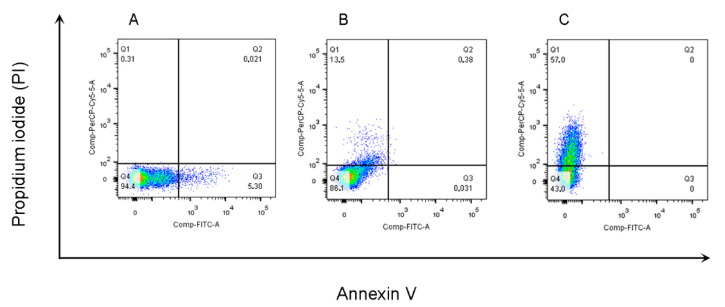
Detection of parasite membrane changes by concurrent staining with annexin V-FITC and PI. Untreated promastigote forms of *L. amazonensis* (**A**) and treated forms after 24 h with the L1 IC_50_ concentration, 1.24 µM (**B**) and 124 µM (**C**), and subsequently stained with annexin V-FITC conjugate and Pin and measured by flow cytometry. Live promastigote forms (Q4) are both annexin V and PI negative. Primary necrotic parasites (Q1) were only stained with PI.

**Figure 5 ijms-23-08553-f005:**
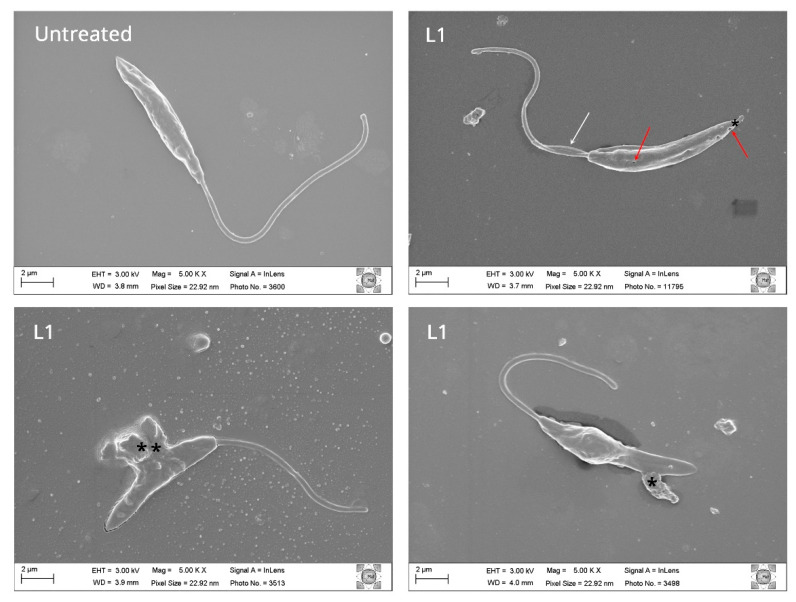
Effect of L1 compound on *Leishmania amazonensis* promastigote. Untreated parasite and promastigote forms treated for 24 h with L1 compound at concentration 1.24 µM were observed by scanning electron microscopy and images were acquired. Treated parasites evidence morphological changes in the membrane with (*) with possible extravasation of cell content (**), cytoskeleton modification by deforming of parasite's flagellum shape (white arrow), and pores (red arrow).

**Figure 6 ijms-23-08553-f006:**
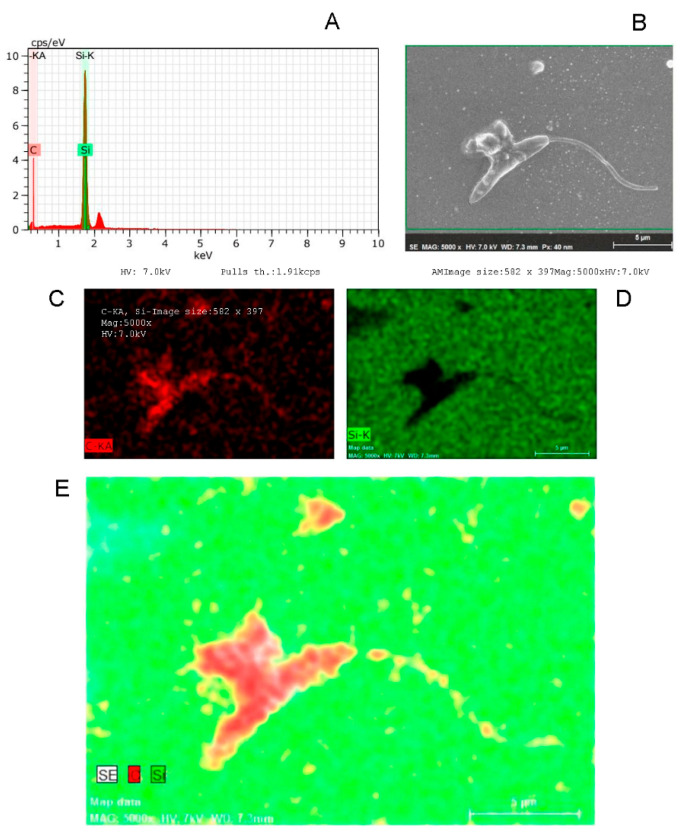
Organic contents of *L. amazonensis* promastigotes treated with L1. SEM images were analyzed by X-ray spectroscopy. The diagram shows the results of Spectroscopy of Scattered Energy (SSE) analysis (**A**); graphical representation of the content in the SEM image (**B**); carbon (**C**, red); silicon (**D**, green) map of the cellular content; and merging (**E**) of the results.

**Figure 7 ijms-23-08553-f007:**
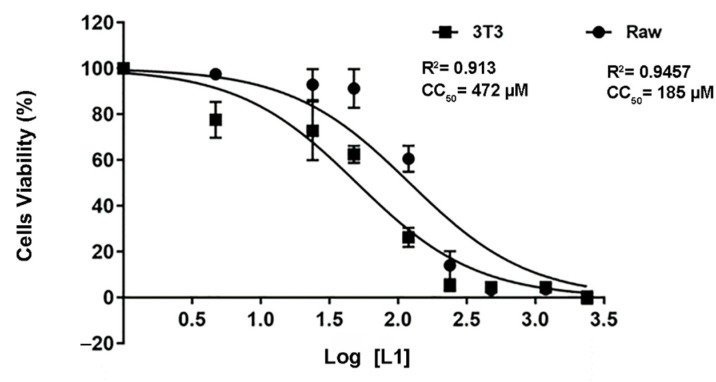
L1 evidence of the low toxicity to macrophage and fibroblast cell lines. The viability of 3T3 (fibroblasts) and Raw (macrophages) cells treated with L1 at different concentrations for 24 h was evaluated by the MTT method. The data are presented as the arithmetic mean and standard deviation (±). ANOVA and Dunnett’s post-test multiple comparisons were applied to determine the statistical significance between the different concentrations tested considering the value of *p* < 0.0001. Results are presented as the mean and standard derivation.

**Table 1 ijms-23-08553-t001:** Analyzed compounds in this study were labeled with Arabic numerals (Zinc database structures), and their structure shown. The highest score estimated the binding free energy (EFE) and the final intermolecular energy (FIE) based on docking of the crystal structure of the glycoprotein (gp63) with ligands, and are shown in the table. AutoDock4 estimates the ligand interactions of the docked complexes with the least global energy.

Ligands	Structures	EFE (kcal/mol)	FIE (kcal/mol)
DETC (ZINC_03633221_)	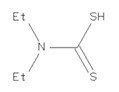	−4.45	−5.64
Doxycycline (ZINC_21984014_)	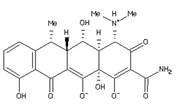	−5.46	−7.54
Terpinen-4-ol (ZINC_03861537_)	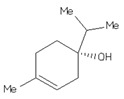	−5.04	−5.64
L1	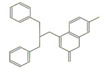	−8.07	−9.86

**Table 2 ijms-23-08553-t002:** Antiparasitic activity of the compounds against *L. amazonensis* promastigotes, as performed by resazurin assay. Promastigotes were treated after 24 h and the IC_50_ was determined. Each value expresses the mean of three experiments performed in triplicate ± standard deviation, using a regression curve.

Compounds	IC_50_ (µM)
AmpB	0.906 ± 0.06
Doxycycline	26.82 ± 0.1
DETC	4.97 ± 0.119
L1	1.24 ± 0.08746

IC_50_—half of the inhibitory concentration.

**Table 3 ijms-23-08553-t003:** Selectivity index values of the L1 compound for the parasite and mammalian cell lines.

Cells	IC_50_—CC_50_ (µM)	SI
*L. amazonensis*	1.24 ± 0.0874	-
3T3	185 ± 0.0524	149.19
RAW	472 ± 0.0706	380.64

Selectivity index (SI): Ratio between CC_50_ and IC_50_.

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
