# Peer review of "Quantum Biochemistry Screening and In Vitro Evaluation of Leishmania Metalloproteinase Inhibitors"

_ijms, 2022, doi:10.3390/ijms23158553_

Round 1
Reviewer 1 Report
This is an interesting manuscript and I have the following suggestions the author may like to consider.
Abstract
A selective index in a number and has no units as it compares the activity against an organism and mammalian cells.
Key words: why does ‘Leishmanolysin’ have a capital letter?
Introduction
Leishmaniasis is caused by infection with a Leishmania parasite. The first sentence needs changing.
Leishmaniasis only has a capital letter when it is the first word in a sentence.
What about mucocutaneous leishmaniasis – this form seems to be ignored. I know some people may think it’s a type of cutaneous leishmaniaisis, but it is different.
What does ‘eco‑epidemiological conditions’ mean?
‘These aspects become the design of control’ – this sentence does not make sense. This whole paragraph needs rewriting as its not logical e.g. ‘high toxic’ is not a sensible phrase. I will not point any other problems with the English.
Can the authors explain how Gp63 is a virulence factor? How does it make the parasite more virulent? You have to explain how the changes caused by this protein allow the parasite to be more infective and not assume the reader makes the connection. What does ‘inside migration’ mean – migrating where? To sites within the body or within host macrophages?
‘proteases have been considered promising targets in protozoa infection treatment’ Why? Were compounds less effective as they didn’t target specific molecules in the host or parasite or they needed to be used at high concentrations? What does your ‘a specific metalloprotease inhibitor molecule’ target? It looks to be as if you did a screen based on modelling and then tested the m ost likely active candidates.
Results
What is a ‘a binding pose’ ?
Do not show data twice. Either have data in the Table and tell us in the text what’s the most active or have the data in the text (Table 2). Did you test your compounds against intracellular macrophages and get an IC50 value? This stage is the more clinically relevant stage. There is no data in Table 3 for a clinically used antileishmanial drug e.g. amphotericin B. Does the data in Table 3 use IC50 data for the amastigotes stage? It is not clear to me. What would be considered a good activity for a novel antileishmanial – nM range?
Discussion
How would you see your novel compound being used in the clinic?
Author Response
Introduction
Leishmaniasis is caused by infection with a Leishmania parasite. The first sentence needs changing.
Answer: The sentence has been updated in the new version of the manuscript.
Leishmaniasis only has a capital letter when it is the first word in a sentence.
Answer: The sentence has been updated in the new version of the manuscript.
What about mucocutaneous leishmaniasis – this form seems to be ignored. I know some people may think it’s a type of cutaneous leishmaniaisis, but it is different.
Answer: The sentence has been updated in the new version of the manuscript.
What does ‘eco‑epidemiological conditions’ mean?
Answer: The sentence has been updated in the new version of the manuscript.
‘These aspects become the design of control’ – this sentence does not make sense. This whole paragraph needs rewriting as its not logical e.g. ‘high toxic’ is not a sensible phrase. I will not point any other problems with the English.
Answer: The sentence has been updated in the new version of the manuscript.
Can the authors explain how Gp63 is a virulence factor? How does it make the parasite more virulent? You have to explain how the changes caused by this protein allow the parasite to be more infective and not assume the reader makes the connection. What does ‘inside migration’ mean – migrating where? To sites within the body or within host macrophages?
Answer: The sentence has been updated in the new version of the manuscript.
‘proteases have been considered promising targets in protozoa infection treatment’ Why? Were compounds less effective as they didn’t target specific molecules in the host or parasite or they needed to be used at high concentrations? What does your ‘a specific metalloprotease inhibitor molecule’ target? It looks to be as if you did a screen based on modelling and then tested the most likely active candidates.
Answer: The sentence has been updated in the new version of the manuscript.
Results
What is a ‘a binding pose’?
Answer: The sentence has been corrected (poses) in the new version of the manuscript.
Do not show data twice. Either have data in the Table and tell us in the text what’s the most active or have the data in the text (Table 2). Did you test your compounds against intracellular macrophages and get an IC50 value?
Answer: No, in the future we will test against amastigote forms. Although that has a lower number of metalloproteases, the protein is exposed to interaction with the host's immune system. Thus, in vivo model should be our priority at the moment.
This stage is the more clinically relevant stage. There is no data in Table 3 for a clinically used antileishmanial drug e.g. amphotericin B.
Answer: No, because it already used in clinical.
Does the data in Table 3 use IC50 data for the amastigotes stage?
Answer: No, not yet. In near future, we intend to proceed with the studies against amastigote and in vivo animal models.
It is not clear to me. What would be considered a good activity for a novel antileishmanial – nM range?
Answer: IC50 for L1 (1.24) against promastigote form was similar to the current drug already used in clinical practice, amphB (0.9). Taking into account the lack of treatment, for us to justify continuing the studies in order to improve its characterization and perhaps become an ally in the fight against Leishmaniasis.
Discussion
How would you see your novel compound being used in the clinic?
Answer: L1 showed characteristics of a promising compound, being able to reduce the proteolytic action of metalloproteases from L. amazonensis on the gelatin substrate (Data not showed). L1 was shown to be a strong protease inhibitor compound, with low cytotoxicity for mammalian cells and high selectivity index. Currently, the drugs used to control leishmaniasis show significant cytotoxicity, in addition to drug resistance cases. We intend to proceed with the studies against amastigotes, although that has a lower number of metalloproteases, the protein is exposed to interaction with the host's immune system. Thus, in vivo model should be our priority at the moment. When we elucidate the cytotoxicity against intracellular forms of amastigotes, in vivo model, can continue the further characterization of the compound as a drug. Our first line of thinking is that L1 can be a combined therapy, being used as a protease target but in association with amph B, as is already done in other infections in which protease plays important roles, such as viruses, for example. On the other end, in our lab we currently work with nanoformulation, may in the future we can use nanotechnology and develop new formulations.
Reviewer 2 Report
Quantum biochemistry of gp 63 protease
This is an interesting and created approach to evaluate this important leishmania enzyme. However, there are so many grammar, syntax, spelling errors, and word choice problems in the text that it is very difficult to read. Some ‘sentences’ are truly not sentences. I have indicated below the many lines in the text with issues: line 55 (why capitalization of leishmaniasis?); 61, 62, 63, 65, 69, 70, 73, 75, 76, 78, 82,84,8592,93,95,105107-108 (not a sentence), 121, 122 (not a sentence); 153-154 (not a sentence); 257, 260, 274 (units of concentration?);line 289 (where was CC50 value defined?);294, 300;305; 315;317;320;332 (bonds or interactions??);333 (is this a sentence?);337,339;341; 342; 343 (is L1 really a Lewis base?); 348; 367 (not a sentence); 371 (what is a filament?);379 (which covalent type of bond??); 390 (not a sentence); 414 (sentence meaning is not clear);458; 499 (cells from what source?);lines 503-504 are not clear; 512; 516 (570 nm for excitation and 600 nm for emission??); 528; 569 (what solvent was used for control cells?); 586; 588; 600-602 this seems to be just ‘stuck onto’ the end and does not follow from previous sentences.
The abstract clearly indicates the work being presented and is nicely written.
I have problems understanding line 109 indicating covalent interactions…what are the authors really saying: that new covalent bonds are made? Has this been established by evaluating these suspected covalent interactions by other methods such as dissociation curves??
Line 343 indicates the compounds can act as Lewis bases; after looking at the L1 structure (Table 1)…where is the Lewis basicity (the unshared electron pair) for L1??
For Table 2, which viability assay was used to obtain these data?
Line 246: why 4.78 uM level of L1 (since the IC50 value was reported in Table 2 as 1.24 uM ?? Same question for lines 525-526.
In Table 3 only L1 data were shown; the authors indicate in the text that the other compounds are in clinical use; however, it would be helpful to indicate some literature data from these compounds to help put the L1 data in Table 3 in context.
For lines 412 and 413 authors should give a reference for this
From line 506: where were the data from the resazurin assay presented?
Author Response
This is an interesting and created approach to evaluate this important leishmania enzyme. However, there are so many grammar, syntax, spelling errors, and word choice problems in the text that it is very difficult to read. Some ‘sentences’ are truly not sentences. I have indicated below the many lines in the text with issues: line 55 (why capitalization of leishmaniasis?) ; 61, 62, 63, 65, 69, 70, 73, 75, 76, 78, 82,84,8592,93,95,105107-108 (not a sentence), 121, 122 (not a sentence) 153-154 (not a sentence); 257, 260, 274 (units of concentration?);line 289 (where was CC50 value defined?.294, 300;305; 315;317;320;332 (bonds or interactions??);333 (is this a sentence? 337,339;341; 342; 343 (is L1 really a Lewis base? 348; 367 (not a sentence) 371 (what is a filament?);379 (which covalent type of bond??) 390 (not a sentence); 414 (sentence meaning is not clear); 458; 499 (cells from what source? );lines 503-504 are not clear; 512; 516 (570 nm for excitation and 600 nm for emission??); 528; 569 (what solvent was used for control cells?) 586; 588; 600-602 this seems to be just ‘stuck onto’ the end and does not follow from previous sentences.
Answer: All sentences have been updated in the new version of the manuscript. Filament is a long chain of ligands. The covalent bonds with the amino acids – alkyl group and hydrogen bonding were observed. DMSO was solvent used for control cells.
The abstract clearly indicates the work being presented and is nicely written.
Answer: Thanks very much for comments.
I have problems understanding line 109 indicating covalent interactions…what are the authors really saying: that new covalent bonds are made?
Answer: Yes, covalent bond type: p-Alkil, hydrogen, and amino acids.
Has this been established by evaluating these suspected covalent interactions by other methods such as dissociation curves??
Answer: No.
Line 343 indicates the compounds can act as Lewis bases; after looking at the L1 structure (Table 1)…where is the Lewis basicity (the unshared electron pair) for L1??
Answer: Yes, the modeling data demonstrated the interaction between L1 and gp63 due to its structure and size.
For Table 2, which viability assay was used to obtain these data?
Answer: Resazurin assay
Line 246: why 4.78 uM level of L1 (since the IC50 value was reported in Table 2 as 1.24 uM ?? Same question for lines 525-526.
Answer: It was a typing error. We performed the IC50 using two methodologies. The first one was based on microscopy, more empirical and less accurate. The count showed an IC50 of 4.78. We also made the resazurin kit assay, which is more specific and accurate with 1.24 of IC50.
In Table 3 only L1 data were shown; the authors indicate in the text that the other compounds are in clinical use; however, it would be helpful to indicate some literature data from these compounds to help put the L1 data in Table 3 in context.
Answer: It is already indicated during the discussion.
For lines 412 and 413 authors should give a reference for this
Answer: The sentence has been updated in the new version of the manuscript.
From line 506: where were the data from the resazurin assay presented?
Answer: These data are presented in table 2.
Round 2
Reviewer 2 Report
While this manuscript is improved in grammar and word choice, I still have some problems with the following lines in the text (since these take away from the message of the manuscript):
Line 47 change ‘exist’ to have
Line 105 change biding to binding
Line 106 change taking to taken
Line 108 change poses to pose and its to their
Line 269 remove ‘of’
Line 305 change was to had
For figure 7, remove ‘de’
Line 317 change media to mean
Line 507 what is the purpose of -DMSO
Lines 516-517 mean is unclear
Line 529 should indicate that the excitation is at 570 nm and the emission at 600 nm (where data were obtained)
Line 538…why use of 4.78 uM
Line 541: change washing to washed
Line 582 remove ‘of’
Line 603 purpose of word ‘scape’ ?
Line 604 appears to not be a sentence
Line 609 change fourth to four
Line 672 change to ‘therapeutic’
In addition, I still struggle to understand the authors’ use of ‘covalent bond’; perhaps they can define this by indicating their criteria such as bond distance of a certain range.
Author Response
Dear Reviewer.
Thank you for your comments and corrections.
All word and sentence changes requested by the reviewer were made in the new version of the manuscript. All changes are highlighted (green) in the new version of the manuscript.
In addition, the use of the term "covalent bonds" was established in the manuscript due to the use of two computational methods, in addition to the distance predicted by the MFCC in combination with (eV) - total binding energy, allowed us to reinforce the evidence of the existence of bonds covalent bonds between compound L1 and protein gp63.
Thank you very much for your contribution to improving the quality of our manuscript.